# Targeting anti-apoptotic pathways eliminates senescent melanocytes and leads to nevi regression

Jaskaren Kohli[1], Chen Ge [1], Eleni Fitsiou [1], Miriam Doepner[2], Simone M. Brandenburg [1], William J. Faller [3], Todd W. Ridky [2] & Marco Demaria [1] ✉

Human melanocytic nevi (moles) result from a brief period of clonal expansion of melanocytes. As a cellular defensive mechanism against oncogene-induced hyperplasia, nevus-resident melanocytes enter a senescent state of stable cell cycle arrest. Senescent melanocytes can persist for months in mice and years in humans with a risk to escape the senescent state and progress to melanoma. The mechanisms providing prolonged survival of senescent melanocytes remain poorly understood. Here, we show that senescent melanocytes in culture and in nevi express high level of the anti-apoptotic BCL-2 family member BCL-W but remain insensitive to the pan-BCL-2 inhibitor ABT-263. We demonstrate that resistance to ABT-263 is driven by mTOR-mediated enhanced translation of another anti-apoptotic member, MCL-1. Strikingly, the combination of ABT-263 and MCL-1 inhibitors results in synthetic lethality to senescent melanocytes, and its topical application sufficient to eliminate nevi in male mice. These data highlight the important role of redundant anti-apoptotic mechanisms for the survival advantage of senescent melanocytes, and the proof-of-concept for a non-invasive combination therapy for nevi removal.

All adult humans have a variable number of melanocytic nevi or moles normally originating at young ages and able to persist for decades. Besides cosmetic reasons, removal of irregular or hyperplastic nevi can serve as a preventative strategy against melanomagenesis as 20–30% of existing melanocytic nevi are able progress to melanoma, although there is debate as to whether this statistic is truly representative[1]. While surgery is the most common and effective strategy for nevi removal, this is complicated for large nevi as it may require a skin graft procedure and promote excessive scarring. An alternative strategy is the use of Q-switched ruby laser (QSRL), a beam of light with wavelength of 694 nm which specifically target melanin. Efficacy of QSRL remains quite controversial as it often promotes only a partial and reversible depigmentation effect, and can even lead to melanomagenesis[2].

Currently, other therapies for the removal of melanocytic nevi, including targeted approaches, are not available.

The majority of human melanocytic nevi (~82%) derives from an activating mutation in the gene *BRAF*, a serine/threonine protein kinase that plays an important role in transducing growth factor signals to the nucleus[3,4]. BRAF mutant nevi are generally much smaller compared to congenital nevi, which are predominately driven by mutations in the NRAS oncogene[5]. Constitutively activated BRAF (*BRAF*[V600E]) causes an initial brief period of clonal expansion of the mutant melanocytes[6]. As a cellular defensive mechanism against oncogene-induced hyperplasia, the tumor suppressor gene encoding the Cyclin Dependent Kinase (CDK)-4/6 inhibitor p16 gets induced, causing an irreversible cell cycle arrest termed oncogene-induced

[1]European Research Institute for the Biology of Ageing, University Medical Center Groningen, Groningen 9713 AV, Netherlands. [2]Department of Dermatology, Perelman School of Medicine, University of Pennsylvania, Philadelphia, PA 19104, USA. [3]Netherlands Cancer Institute, Amsterdam 1066 CX, Netherlands. ✉e-mail: m.demaria@umcg.nl

senescence (OIS). In nevi, p16 expression is mosaic, consisting of a heterogenous population of nevus cells expressing either high or undetectable levels of p16[7,8]. Besides p16 expression, senescent cells also develop a complex senescence-associated secretory phenotype (SASP) that entails the expression and secretion of numerous pro-inflammatory factors[9,10]. Pro-inflammatory SASP factors can serve as activator of immunosurveillance mechanisms and help to restrict the persistence of senescent cells in tissues via immune-mediated clearance[11]. In the skin, stromal senescent cells are recognized and cleared by NK and CD8+ T cells[12]. However, nevus-resident senescent melanocytes persist for months in mice and years in humans. Many senescent cells are primed to apoptose but remain alive because of the upregulation of various anti-apoptotic proteins. Drugs interfering with such apoptosis-resistance mechanisms have been proven to be selectively toxic against senescent cells and have led to the development of a novel class of compounds termed senolytics[13].

Because a well-conserved anti-apoptotic mechanism during senescence is the overexpression of anti-apoptotic members of the BCL-2 family[11], potent senolytics are inhibitors of the BCL-2- family such as ABT-263 and ABT-737[14,15]. However, responses to BCL-2 inhibitors and other senolytics remain variable, mainly because of the high heterogeneity observed in different types of senescence. Here, we study the expression and modulation of BCL-2 family members in human and mouse senescent melanocytes with the goal to identify a novel targeted and non-invasive therapy for the removal of melanocytic nevi in vivo.

## Results

### Senescent melanocytes are resistant to BH3 mimetics despite upregulation of BCL-w

To determine the expression of anti-apoptotic members of the Bcl-2 family in senescent melanocytes, we initially used a model of irradiation-induced senescence. Primary human melanocytes were exposed once to a 10 Gy dose of γ-radiation (IR), and mock-radiated cells were used as controls. Senescence induction was confirmed in IR-treated, but not mock-treated, melanocytes by enhanced activity of the senescence-associated β-galactosidase (SA-β-gal), reduced number of actively proliferating (EdU+) cells, and increased expression of the *p16*, *p21*, and of the SASP factors *IL6* and *MMP1* (Fig. 1a–c). Among anti-apoptotic Bcl-2 family members we observed a significant upregulation of BCL-w in senescent melanocytes, while no differences were observed for BCL-2 and BCL-xl at both mRNA (Fig. 1d) and protein level (Fig. 1e). We also observed a significant decrease in Mcl-1 levels (supplementary fig. 1a). A previous report from our group have demonstrated that elevated BCL-w level is a conserved feature in senescence[16]. While no specific inhibitors of BCL-w exist, senescent cells are sensitive to the BCL inhibitors ABT-263 and ABT-737, which target BCL-w, BCL-2, and BCL-xL[14,17]. However, senescent melanocytes were minimally sensitive to ABT-263 (Fig. 1f) and ABT-737 (supplementary fig. 1b). We decided to also test whether similar results would be obtained in BRAFV600E-induced senescent melanocytes. Senescence was confirmed in this experimental model by enhanced numbers of SA-β-gal positive cells, reduced numbers of EdU+ cells, and expression of *p16* and *p21* (Fig. 1g–i). BRAF[V600E]-induced senescent melanocytes also significantly upregulated *Bcl-w* mRNA expression (Fig. 1j). To verify altered expression of BCL-w in senescent melanocytes in vivo, we then measured its levels in human skin and compared melanocytes populating nevi, which are primarily senescent[7,8], to melanocytes populating the epidermis, which are mainly non-senescent[7,8]. Interestingly, the number of Melan-a+ cells expressing BCL-w was significantly higher in the nevi compared to the epidermis (Fig. 1k, l). In contrast, the expression of other Bcl-2 members was similar between nevi and epidermis (supplementary fig. 1c). As for irradiated melanocytes, BRAF[V600E]-induced senescent melanocytes were resistant to ABT-263 (Fig. 1m), in particular when compared to

primary IMR-90 fibroblasts where the treatment achieved ~70% toxicity (Fig. 1n). In accordance to unchanged levels of BCL-2 and BCL-xL, senescent melanocytes were also completely insensitive to BCL-xL- and BCL-2-specific inhibitors A-1155463 (Fig. 1o) and ABT-199 (Fig. 1p), previously shown to be toxic for certain senescence subtypes[18,19]. Cells sensitive to pan-BCL-2 inhibition, in particular cancer cells, are normally primed to apoptose due to elevated levels of the pro-apoptotic factor Noxa[20,21]. However, irradiated and BRAF[V600E] senescent melanocytes also expressed higher levels of *Noxa* mRNA (Fig. 1q, r) and protein (Fig. 1s) compared to non-senescent counterparts. Altogether, these data suggest that despite elevated expression of the anti-apoptotic factor BCL-w and overexpression of the pro-apoptotic factor Noxa, senescent melanocytes remain insensitive to the pan-BCL-2 inhibitors ABT-263 and ABT-737.

### Translational upregulation of MCL-1 mediates resistance to ABT-263

Cancer cells can develop resistance to BCL-2 inhibitors by increasing the expression of another anti-apoptotic protein, MCL-1[22–26]. Interestingly, MCL-1 protein levels were increased in senescent melanocytes at different time points after exposure to ABT-263 (6, 12, and 24 h post treatment) as revealed by western blot (Fig. 2a) and immunofluorescence staining (Fig. 2b, c and supplementary fig. 2a). Previous reports suggested that induction of MCL-1 in resistant cancer cells is due to either enhanced transcription or protein stability. However, neither *Mcl-1* mRNA levels (supplementary fig. 2b) nor protein degradation rates (Fig. 2d) were altered in senescent melanocytes upon treatment with ABT-263. We then decided to evaluate the rate of translation by measuring the abundance of MCL-1 bound to monosome, light polysome, and heavy polysome ribosomal fractions. A significant decrease in the percentage of *Mcl-1* mRNA in the monosome fraction and a concomitant significant increase in the heavy polysome fraction were observed in ABT-263 treated cells (Fig. 2e). To further corroborate these findings, we then labeled nascent proteins with the amino acid analog L-azidohomoalaine (AHA) in senescent melanocytes exposed to either ABT-263 or vehicle. An increased fraction of labeled MCL-1 in cells treated with ABT-263, supporting upregulation of MCL-1 translation, was observed (Fig. 2f).

Previous reports have shown that MCL-1 protein expression is regulated by cap-dependent translation[27,28]. To study whether this was the case in our model, we treated senescent melanocytes with 4EGI-1 which functions by disrupting the association between eIF4E and eIF4G[29]. We observed no increase in ABT-263-mediated MCL-1 levels when cells were co-treated with 4EGI-1 (Fig. 2g), indicating the direct role of cap-dependent translation for the ABT-263-mediated MCL-1 upregulation. As a consequence, treatment with 4EGI-1 sensitized senescent melanocytes to ABT-263, while non-senescent melanocytes remained predominantly viable (Fig. 2h). Because a major mediator of Cap-dependent translation is mTOR, we then tested whether mTOR inhibition could also sensitize senescent melanocytes to ABT-263. We exposed senescent melanocytes to the mTOR inhibitor PP242[30]. The treatment caused the expected inhibition of mTOR activity, as measured by reduced phosphorylation of the mTOR target p70S6K, and also a significant reduction in MCL-1 protein levels (Fig. 2i). Strikingly, treatment with PP242 also sensitized the cells to ABT-263 leading 75% of senescent melanocytes to undergo apoptosis when exposed to a combination of the two drugs (Fig. 2j, supplementary fig. 2c). To validate that Mcl-1 repression by PP242 was responsible for the acquired sensitivity to ABT-263, we repeated the experiment using melanocytes overexpressing MCL-1 (supplementary fig. 2d). When exposed to PP242 and ABT-263, senescent melanocytes overexpressing MCL-1 showed significant resistance to the combination treatment (Fig. 2k). Overall, these results indicate that mTOR-mediated MCL-1 translation is enhanced by the treatment with ABT-263, and that inhibiting mTOR activity sensitizes senescent melanocytes to ABT-263.

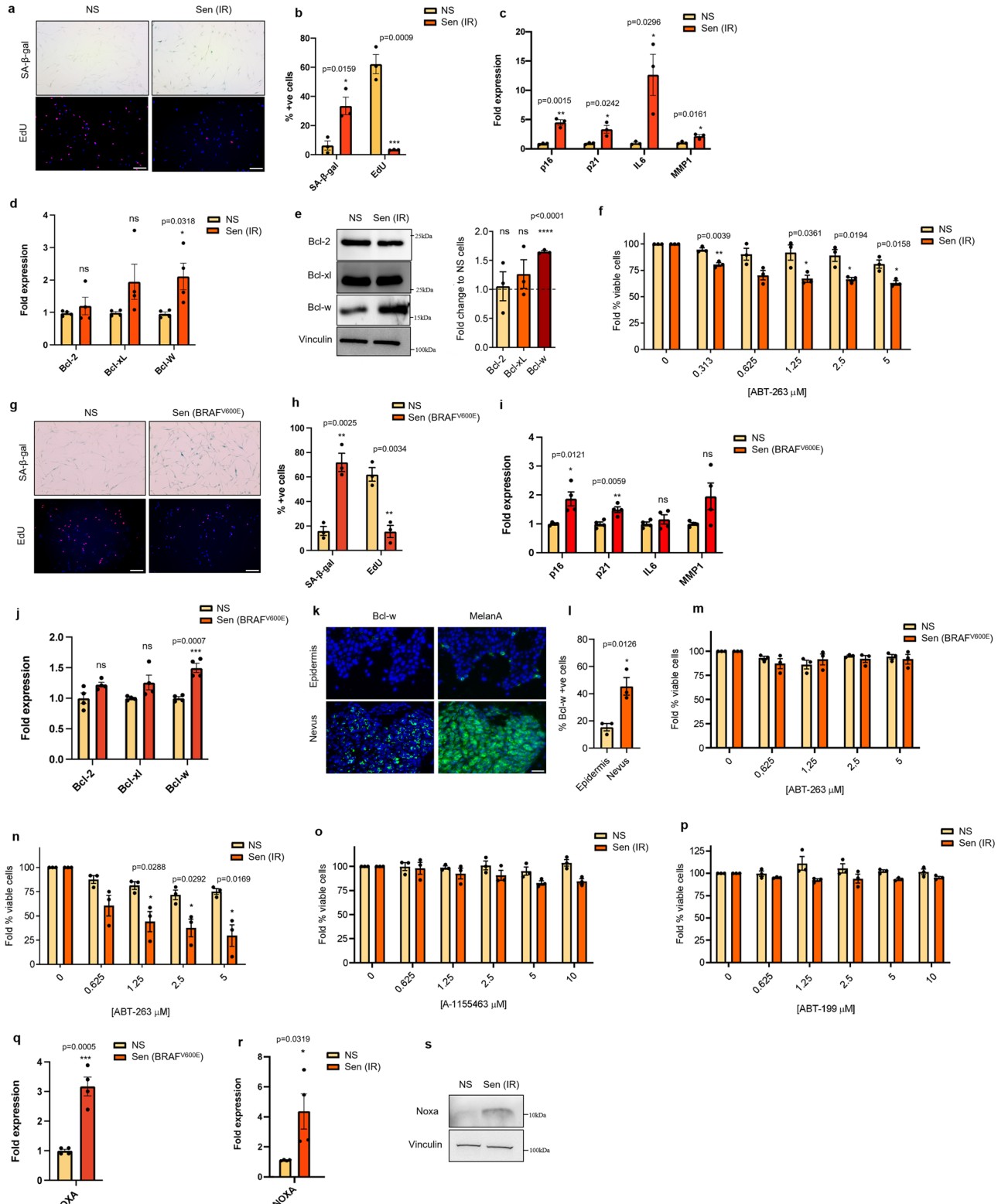

## ABT-263 synergizes with MCL-1 inhibition to induce apoptosis in senescent melanocytes

mTOR is a general regulator of protein translation and regulates the expression of hundreds of proteins. Thus, its inhibition affects various signaling pathways and might not represent a valuable therapeutic strategy to specifically target the resistance of senescent melanocytes to ABT-263. In contrast, our data suggested that MCL-1 might be the mediator of the resistance and could potentially be a more specific target. Exposure to small interfering RNA (siRNA) against *Mcl-1* caused ~70% cell death in senescent melanocytes treated with ABT-263, while no effect was observed in non-senescent cells (Fig. 3a, b, supplementary fig. 3a). Next, we tested if similar sensitization to ABT-263 could be achieved by exposing cells to pharmacological MCL-1 inhibitors. Co-treatment with ABT-263 and MCL-1 inhibitors S63845[31] or A-1210477[22] resulted in a dramatic reduction in viability of senescent melanocytes 10 and 20 days after irradiation (Fig. 3c−e and supplementary fig. 3b).

**Fig. 1 | Senescent melanocytes are resistant to BH3 mimetics despite upregulation of BCL-w and NOXA. a** SA-β-gal/EdU images and **b** quantification of non-irradiated (NS) and irradiated senescent (Sen IR) melanocytes. Scale bar = 100 μM. **c** mRNA analysis of *p16*, *p21*, *IL6*, and *MMP1*. Expression was normalized using Cq values of *tubulin*. **d** mRNA expression of *Bcl-2*, *Bcl-xL*, and *Bcl-w* from irradiated melanocytes. Expression was normalized using Cq values of *tubulin*. **e** Protein expression and densitometry analysis of BCL-2, Bcl-xl, and Bcl-w from irradiated melanocytes. Vinculin was used as a housekeeping protein and expression was normalized to protein levels in non-irradiated non-senescent cells. **f** Viability of non-irradiated and irradiated melanocytes treated with indicated concentrations of ABT-263 for 24 h. **g** SA-β-gal/EdU images and **h** quantification of puromycin vector transduced (NS) and BRAF^V600E vector transduced senescent (Sen BRAF^V600E) melanocytes. Scale bar = 100 μM. **i** mRNA analysis of *p16*, *p21*, *IL6*, and *MMP1*. Expression was normalized using Cq values of *tubulin*. **j** mRNA expression of *Bcl-2*, *Bcl-xL*, and *Bcl-w* from irradiated melanocytes. Expression was normalized using Cq values of *tubulin*. **k** Immunofluorescence and **l** quantification of Bcl-w positive cells in human nevi. Melan-a was used to confirm the identification of melanocytes or nevus cells. Scale bar = 25 μM. **m** Viability of non-senescent and BRAF^V600E transduced melanocytes treated with indicated concentrations of ABT-263 for 24 h. **n** Viability of non-irradiated and irradiated IMR-90 fibroblasts treated with indicated concentrations of ABT-263 for 24 h. Viability of non-irradiated and irradiated melanocytes treated with indicated concentrations of **o** A-1155463 or **p** ABT-199 for 24 h. mRNA analysis of *NOXA* in **q** BRAF^V600E transduced and **r** irradiated melanocytes. Expression was normalized using Cq values of *tubulin*. **s** Protein expression of NOXA from irradiated melanocytes. Vinculin was used as a housekeeping protein. For all panels, *n* = 3 independent experiments except **d**, **i**, **j**, and **q** where *n* = 4 independent experiments, and **s** where results are representative of *n* = 2 independent experiments. Significance was calculated with a two-tailed student's *t* test. Error bars = S.E.M. *\*p* < 0.05, *\*\*p* < 0.01 *\*\*\*p* < 0.001, ns = not significant.

In contrast, no decrease in viability was observed on proliferating or senescent melanocytes when treated with MCL-1 inhibitors as lone agents (Fig. 3f and supplementary fig. 3c). To determine whether the toxic effect of ABT-263 and MCL-1 inhibitors against senescent melanocytes was achieved via induction of apoptosis, we measured the level of Annexin V/PI+ cells. We observed a drastic decrease in the percentage of Annexin V/PI viable senescent melanocytes (-75%) when both drugs were used (Fig. 3g and supplementary fig. 3d). In contrast, only a small decrease (≤15%) was seen in non-senescent cells. Similarly, caspase 3/7 activity was significantly increased in senescent cells treated with the combination, while no effect was observed in proliferating cells or in senescent cells treated with the individual agents (Fig. 3h and supplementary fig. 3e). Next, we exposed senescent melanocytes to ABT-263 and MCL-1 inhibitors in the presence of the pan-caspase inhibitor QVD. Treatment with QVD completely rescued senescent melanocytes from cell death, further validating that exposure to ABT-263 and MCL-1 inhibitors led senescent melanocytes to caspase-dependent intrinsic apoptosis (Fig. 3i and supplementary fig. 3f). Interestingly there was an increase in viability in cells treated with QVD and both ABT-263 and Mcl-1 inhibitors compared to QVD only treated cells. However, surviving cells were still arrested. We then tested the effect of combinatorial treatment in different models of senescent melanocytes. Induction of senescence in melanocytes exposed to etoposide and UV induced senescence was confirmed by enhanced SA-β-gal activity, reduced number of EdU+ cells, and upregulation of cell cycle arrest and SASP factors (supplementary fig. 3g–l). Importantly, the toxic effects of the combination ABT-263 and MCL-1 inhibitors were observed in all senescent models, while lone treatment with ABT-263 exerted little effect (Fig. 3j–m). Interestingly, treatment with MCL-1 inhibitors did not sensitize ABT-263-resistant BJ fibroblasts (supplementary fig. 3m, n), despite the treatment with ABT-263 resulted sufficient in promoting MCL-1 expression (supplementary fig. 3o). Overall, these results show that MCL-1 inhibition specifically sensitize ABT-263-treated senescent melanocytes to undergo apoptosis, independent from the senescent stimulus.

## ABT-263 and S63845 can induce death in nevi in mice and human skin organoids

Because nevi are populated by senescent-like melanocytes, we reasoned that the combination of ABT-263 and MCL-1 inhibition could potentially lead to nevus reduction by inducing targeted cell death. To test this hypothesis, we used an inducible mouse model where melanocyte-specific expression of BRAF^V600E upon hydroxytamoxifen (4-OHT) administration induces the generation of senescent pigmented nevi[32–35]. The schematics for this experiment are illustrated in Fig. 4a. Tamoxifen was topically applied on the skin of 9-week old mice, with pigmented nevi developed within 2 months, typically localized to the dermis (Fig. 4b) as reported elsewhere[33,36,37]. mRNA analysis displayed a significant increase and a positive correlation in the

transcription of the senescent marker *p16* and of the melanocyte marker *SOX10* in nevi compared to skin (Fig. 4c, d). To avoid any potential side effect of systemic administration and to improve drug delivery to the nevus, we topically applied ABT-263, S63845, or the combination of the two onto a small defined area of pigmented mouse skin over a period of 14 days, after which skin was excised and examined for nevus density[38]. ABT-263 and S63845 could be detected by mass spectrometry after topical application on mouse skin (supplementary fig. 4a–c), indicating both drugs do permeate the mouse skin epithelial barrier and reach nevus cells. Strikingly, we observed a significant reduction in nevus density when both ABT-263 and S63845 were co-administered onto mouse skin compared to vehicle control or mice treated with single agents (Fig. 4e, f). Importantly we could still observe pigmented melanocytes in the hair follicles of ABT-263/S63845 treated mice (supplementary fig. 4d), indicating these drugs do not kill quiescent melanocytes. These results indicate that nevi-resident melanocytes require pan inhibition of BCL-2 and MCL-1 anti-apoptotic programs to initiate cell death, and this strategy can be carried out non-invasively on skin via topical administrations.

Finally, we aimed to test whether ABT-263 and S63845 can eliminate senescent melanocytes in a clinically-relevant human system. To do this, we used a human-engineered organotypic skin model containing BRAF^V600E expressing senescent melanocytes[39]. Confirmation of BRAF^V600E expression in melanocytes was carried out by immunoblotting (Fig. 4g). Treating the organoytic cultures with ABT-263 and S63845 resulted in a dramatic reduction in the MelanA-positive area and number of MelanA-positive cells (Fig. 4g–i), without any additional perturbation to the structure. These data indicate that also in clinically-relevant human models the combination of ABT-263 and S63845 can selectively eliminate senescent melanocytes.

## Discussion

This study demonstrates that senescent melanocytes in culture and in nevi overexpress the anti-apoptotic BCL-2 family member BCL-w, but yet remain resistant to the toxic effect of ABT-263 or ABT-737. These results are surprising considering BCL-w has been previously reported to mediate the survival of senescent fibroblasts that are sensitive to the pro-apoptotic effect of pan BCL-2 inhibition[14]. In contrast to our findings, it has been recently demonstrated that ABT-737 can effectively eliminate senescent melanocytes in 3D skin models[40]. This study did not investigate the effect of ABT-737 on senescent melanocytes neither in 2D cell cultures nor in nevi, and only used the number of p16-positive cells as a readout for treatment efficacy[40]. It is possible that ABT-737 (and possibly other senolytics) efficacy depends on culture conditions or has a selective bias towards eliminating p16-positive cells. Further experiments are warranted to investigate these questions.

Similar to what we observe for senescent melanocytes, various cancer cells upregulate the expression of anti-apoptotic BCL-2 family

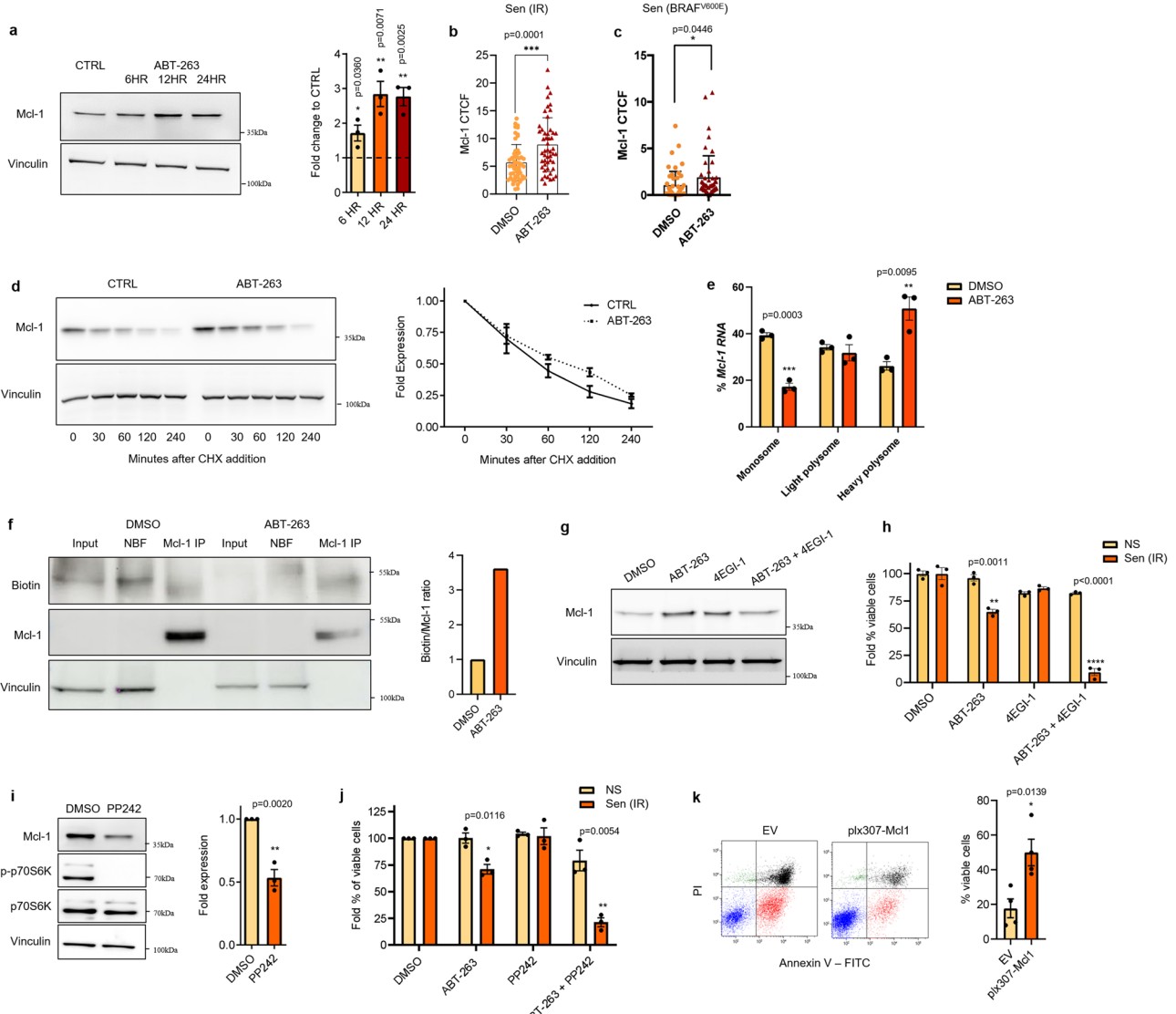

**Fig. 2 | Mcl-1 translation meditates resistance to ABT-263 in senescent melanocytes, which can be repressed through mTOR inhibition. a** Protein expression and densitometry analysis of MCL-1 in irradiated melanocytes treated with 5 μM ABT-263 after 6, 12, or 24 h. Corrected total cell fluorescence of MCL-1 in senescent **b** irradiated and **c** BRAF[V600E] expressing melanocytes treated for 24 h with DMSO or ABT-263. **d** Cycloheximide chase experiment of MCL-1 in DMSO and ABT-263 treated irradiated melanocytes. Cells were incubated with DMSO or 5 μM ABT-263 for 12 h, after which 5 μg/ml cycloheximide was added. Protein lysates were obtained after the indicated time points and used for Mcl-1 immunoblotting. Quantification of MCL-1 degradation rates was carried out using densitometry analysis with vinculin as a housekeeping protein. Expression was normalized to protein levels at time point 0. **e** Distribution of *Mcl-1* mRNA in monosome, light polysome, or heavy polysome fractions from irradiated melanocytes treated with DMSO or 5 μM ABT-263 for 16 h. Distribution of β-actin mRNA was used to normalize values. **f** Immunoblot of immunoprecipitated MCL-1 in vehicle or ABT-263 treated irradiated melanocytes showing nascent (biotinylated) MCL-1 synthesis.

NBT nitro-blue-tetrazalium. Ratio of biotinylated (nascent) MCL-1 protein to total MCL-1 protein in MCL-1 IP fraction. **g** Protein expression of MCL-1 in irradiated melanocytes treated with DMSO, 5 μM ABT-263, and/or 50 μM 4EGI-1. **h** Viability of non-irradiated and irradiated melanocytes treated with 5 μM ABT-263 and/or 50 μM 4EGI-1. **i** Protein expression and densitometry analysis of MCL-1 in DMSO or 1 μM PP242 treated irradiated melanocytes after 24 h. Vinculin was used as a housekeeping protein and expression was normalized to protein levels in DMSO treated cells. **j** Quantification of Annexin V/Pi flow cytometric analysis of non-senescent (NS) and senescent (Sen IR) melanocytes treated with DMSO, 5 μM ABT-263, 1 μM PP242, or 2.5 μM ABT-263/1 μM PP242 for 24 h (see supplementary fig. 2c). The percentage of viable cells in drug-treated cells were normalized to the percentage of DMSO treated viable cells. **k** Annexin V/Pi flow cytometric analysis and quantification of EV or plx307-Mcl1 transduced irradiated melanocytes treated with 2.5 μM ABT-263/1 μM PP242 for 24 h. For all panels significance was calculated with a two-tailed student's *t* test and *n* = 3 independent experiments except **k** where *n* = 4. Error bars = S.E.M. **p* < 0.05, ***p* < 0.01, ****p* < 0.001, ns = not significant.

members but yet remain insensitive to ABT-263 or ABT-737. Interestingly, the combination of BCL-2 and MCL-1 inhibitors has been shown to be selectively toxic melanoma[24,25,41,42], acute myeloid leukemia[43], and cervical cancer cells[22,23], but mechanisms for this synergy remain elusive. We find that senescent melanocytes translationally upregulate MCL-1 in response to ABT-263 as a mechanism to evade ABT-263-mediated apoptosis. This phenomenon is mainly regulated by mTOR, and the mTOR inhibitor PP242 is sufficient to decrease MCL-1 levels

and sensitize senescent melanocytes to the toxic effect of ABT-263. Although a similar mechanism has been observed in small-cell lung cancer cells[44], to our knowledge this is the first study demonstrating this occurs in senescent primary cells. These results may also suggest that mTOR inhibition could be useful in other physiological contexts where cells are resistant to ABT-263. Chemotherapeutic agents such as doxorubicin induce senescence[45] and rapamycin has been reported to synergize with doxorubicin to kill lymphoma cells[28]. mTOR inhibitors

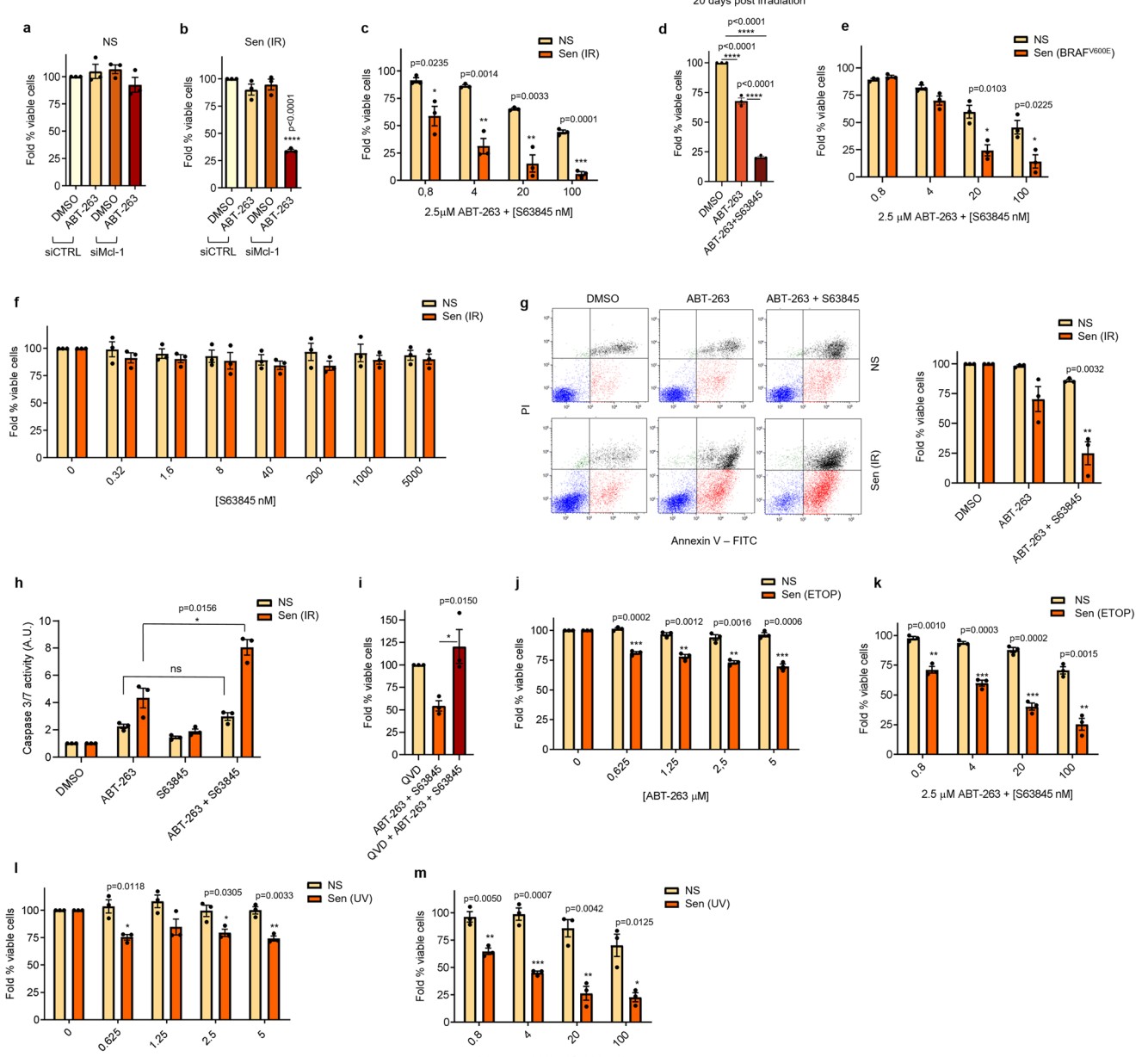

**Fig. 3 | ABT-263 synergizes with MCL-1 inhibition to induce apoptosis in senescent melanocytes. a** Viability of non-irradiated (NS) and **b** irradiated (Sen IR) senescent melanocytes transfected with siCTRL or siMcl-1 and treated with DMSO or 5 µM ABT-263 for 24 h. Viability was normalized to siCTRL cells treated with DMSO. Viability of non-irradiated and **c** 10 day or **d** 20 day irradiated melanocytes treated with indicated concentrations of ABT-263 and S63845 for 24 h. Viability was normalized to untreated cells. **e** Viability of puromycin vector transduced or BRAF^V600E vector transduced cells treated with indicated concentrations of ABT-263 and S63845 for 24 h. **f** Viability of non-irradiated and irradiated melanocytes treated with indicated concentrations of S63845 for 24 h. **g** Annexin V/Pi flow cytometric analysis and quantification of non-irradiated non-senescent and irradiated senescent melanocytes treated with DMSO, 5 µM ABT-263, or 2.5 µM ABT-263/20 nM

S63845 for 24 h. The percentage of viable cells in drug-treated cells was normalized to the percentage of DMSO-treated viable cells. **h** Caspase 3/7 activity in non-irradiated and irradiated melanocytes treated with DMSO, 5 µM ABT-263, 5 µM S63845, or 2.5 µM ABT-263/20 nM S63845 for 24 h. **i** Viability of irradiated melanocytes treated with 20 µM QVD, 2.5 µM ABT-263/4 nM S63845, or 20 µM QVD/ 2.5 µM ABT-263/4 nM S63845 for 24 h. Viability was normalized to untreated cells. Viability of senescent melanocytes induced via etoposide (**j**, **k**) or UV-irradiation (**l**, **m**) treated with indicated concentrations of ABT-263 and/or S63845 for 24 h. Viability was normalized to untreated cells. Significance was calculated using a two-tailed student's *t* test in **c**, **e**, **g**, **h**, **j**–**m**. A one-way ANOVA with Tukey's multiple comparison's test was used in **a**, **b**, **d**, and **i**. For all panels, *n* = 3 independent experiments. Error bars = S.E.M. \**p* < 0.05, \*\**p* < 0.01, \*\*\**p* < 0.001, \*\*\*\**p* < 0.0001.

are also reported to synergize with methotrexate to kill leukemia cells[46], 5-flourouracil to kill gastric cancer cells[47] and cisplatin to eliminate mesothelioma cells[48]. Considering that different mTOR inhibitors are routinely used in the clinic, it can be highly relevant to study whether these reported synergies occurs due to an mTOR-mediated translational decrease in MCL-1.

In alternative to the use of mTOR inhibition, we also demonstrate that pharmacological inhibitors of MCL-1, including S63845, are also able to synergize with ABT-263 to selectively eliminate senescent melanocytes. A recent study found that the combination of ABT-263 and S63845 is similarly synergistically toxic for some senescent breast cancer cells expressing in dependence on NOXA expression[21].

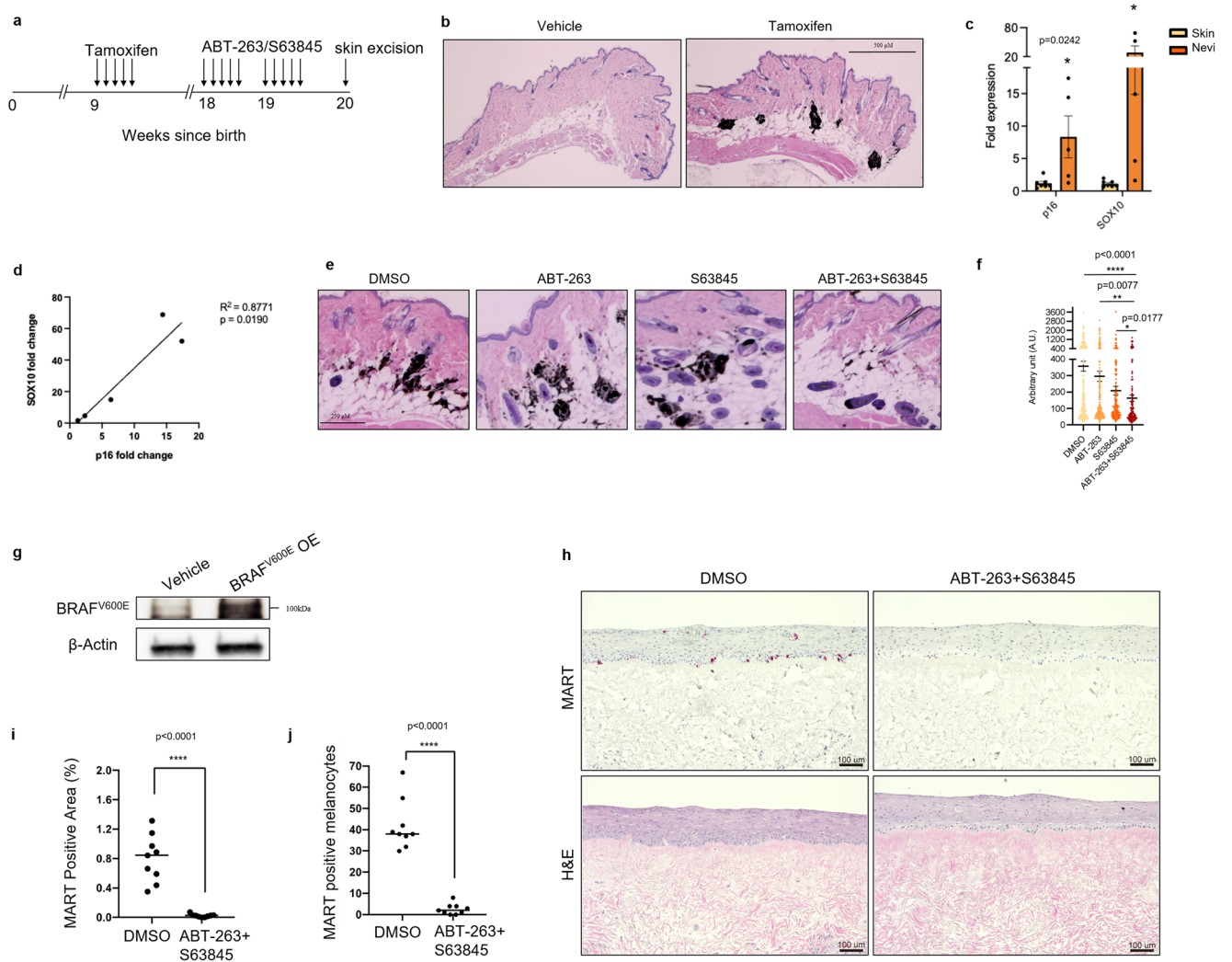

**Fig. 4 | ABT-263 and S63845 can induce death in mouse nevi and human skin organoids. a** Schematic diagram of in vivo experiments in this study. **b** H and E section of nevi in tamoxifen-treated mice. Images are representative of $N = 10$ independent sections. Scale bar = 500 μM. **c** mRNA analysis of *p16* and *SOX10* expression between skin ($n = 7$) and nevi ($n = 5$). Error bars = S.E.M. **d** Scatter analysis of *p16* and *SOX10* in nevi. **e** H and E sections of mouse skin in treated mice. Images are representative of $N = 10$ independent sections. Scale bar = 250 μM. **f** Nevus density quantification of DMSO ($n = 4$), ABT-263 ($n = 4$), S63845 ($n = 3$) and ABT-263/S63845 ($n = 4$) treated mice. **g** Western blot of BRAF$^{V600E}$ overexpression (OE) in melanocytes used for organotypic cultures as in **h**–**j**. The image is representative of three independent experiments. **h** Histological characterization of representative organotypic skin with melanocytes overexpressing BRAF$^{V600E}$, including melanocyte marker MART, and H&E. Scale bar = 100 μM, $n = 3$. **i** Quantification of positive epidermal MART staining area compared to total epidermal area in DMSO and drug-treated organotypics. **j** Quantification of MART positive melanocytes in epidermis via ImageJ. Quantification of nine areas across three biological replicates for each group. Statistical significance in **c**, **i**, and **j** was calculated with a two-tailed student's *t* test, in **d** with a *t* test for linear regression and **f** using a Kruskal-Wallis test with Dunn's multiple comparisons test. *$p < 0.05$, **$p < 0.01$, ****$p < 0.0001$.

However, this study did not investigate whether MCL-1 upregulation due to enhanced translation rates occurs in response to ABT-263.

We show that topical applications of ABT-263 and S63845 induce nevus cell death in vivo. This potential therapeutic strategy could be exploited for the development of non-invasive methods for elimination of melanocytic nevi and reduce the prevalence of melanomagenesis. Although systemic administrations of ABT-263 are reported to induce senolysis in vivo[15,49–51], their use in a clinical setting is hampered from BCL-xL inhibition in platelets, leading to onset of thrombocytopenia[52]. As BCL-w was observed to be engaged in human nevi, it is possible that topical applications of ABT-263 and S63845 could also eliminate senescent human nevus cells. This preventative approach could have implications in cosmetic use for nevus removal, but also be beneficial in reducing the number of melanoma cases known to arise from pre-existing nevi. However, it is also important to consider that this strategy could promote melanoma. p16 positive cells

are heterogeneous in melanocytic nevi[8] and it is unclear if p16-negative cells are non-senescent, or senescent but arrested via unknown mechanisms. If the former, p16 positive senescent cells may suppress the proliferation of p16 negative cells via SASP factors and elimination of p16 positive cells may consequently allow the re-proliferation of p16 negative cells, resulting in melanoma development. Future long-term studies using more faithful mouse models of melanoma are necessary to answer this. Importantly, since senescent melanocytes in old human skin are reported to disrupt tissue architecture[40], it is possible that topical application of senolytics could also be used for skin rejuvenation.

## Methods
### Cell culture
Normal human melanocytes were obtained from the ATCC and maintained in RPMI supplemented with 2 mM L-glutamine, 100 units/ml

penicillin, 100 μg/ml streptomycin, 10% fetal bovine serum, 200 nM 12-O-tetradecanoyl phorbol 13-acetate, 200 pM cholera toxin, 10 nM endothelin 1 and 10 ng/ml human stem cell factor (hereby referred to as melanocyte medium). BJ and IMR-90 fibroblasts were obtained from the ATCC and maintained in complete DMEM (DMEM supplemented with 2 mM L-glutamine, 100 units/ml penicillin, 100 μg/ml streptomycin, and 10% FBS). These cells were maintained in a 5% $O_2$, 5% $CO_2$, 37 °C incubator. 293FT cells were maintained in complete DMEM and maintained in atmospheric oxygen concentrations in a 5% $CO_2$, 37 °C incubator. For cells grown in organotypic skin models, primary human melanocytes and keratinocytes were extracted from fresh discarded human foreskin and surgical specimens as previously described[39,53,54]. The human primary cells were obtained from the University of Pennsylvania Skin Biology and Disease Research Core, which establishes the cultures from deidentified discarded tissues under an approved IRB exempt protocol. Keratinocytes were cultured in a 1:1 mixture of Gibco Keratinocytes-SFM medium + L-glutamine + EGF + BPE and Gibco Cascade Biologics 154 medium with 1% penicillin-streptomycin (ThermoFisher Scientific. # 15140122). Primary melanocytes were cultured in Medium 254 (ThermoFisher, #M254500) with 1% penicillin-streptomycin.

## Plasmids, lentiviral generation, and transduction

BRAF[V600E]-induced senescence was carried out using the HIV-CS-CG-BRAF[V600E]-puro plasmid, or HIV-CS-CG-puro plasmid for control cells[8]. Mcl-1 overexpression was carried out using the plx307-MCL1 plasmid (Addgene # 117726). The plx307 empty vector plasmid was generated by excising the MCL1 open reading frame with NheI (NEB) and SpeI (NEB) enzymes for 1 h at 37 °C. Digested products were resolved on a 0.7% agarose gel and the backbone vector was purified using the Wizard® SV Gel and PCR Clean-Up System (Promega) according to the manufacturer's instructions. Blunt ends were generated using the Quick Blunting Kit (NEB) and subsequently ligated using T4 DNA ligase (NEB) overnight at 16 °C. Ligated vectors were transformed into NEB Stable Competent *E. coli* (C3040), before plating onto ampicillin-resistant plates overnight at 37 °C. A single colony was picked and incubated in liquid LB supplemented with ampicillin overnight at 37 °C. plx307 empty vector plasmid was purified using the PureLink™ HiPure Plasmid Midiprep Kit (ThermoFisher Scientific).

3 μg of plasmids were transfected into 5 million 293FT cells with 3 μg each of viral plasmids (Addgene #12251, #12253, #12259) and Polyfect transfection reagent (Qiagen) in Opti-MEM™ I Reduced Serum Medium (ThermoFisher Scientific). Medium was changed to normal DMEM medium but without antibiotics one day later. 48 h later, medium was spun at 300 g for 5 min and viral particles were concentrated using PEG-it virus precipitation solution (System Biosciences) according to the manufacturer's instructions.

Concentrated virus was resuspended in 200 μl PBS and added onto melanocytes in melanocyte medium (1:50). Viral particles were removed 24 h later and cells were replenished with fresh melanocyte medium. Puromycin selection commenced 24 h later by addition of 1 μg/ml puromycin. For infection with HIV-CS-CG-BRAF[V600E]-puro or HIV-CS-CG-puro plasmids, 6 μg/ml polybrene (Santa Cruz Biotechnology) was also added.

## Senescence induction

γ-irradiation-induced senescence was carried out by subjecting cells to a 10 Gy dose using a Cesium[137] source. For etoposide-induced senescence, melanocytes were cultured with 1 μM etoposide for 3 days. For UV-induced senescence, medium was removed from cells and replaced with PBS. Cells were then exposed to 20 mJ/cm$^2$ UVB irradiation at days 1 and day 3 after plating. Experiments were carried out at day 10 for these types of senescence induction. For BRAF[V600E] induced senescence, melanocytes were infected with HIV-CS-CG-BRAF[V600E]-puro, or

HIV-CS-CG-puro for control cells. Experiments were carried out 14 days post infection.

## SA-β-galactosidase assay

Cells were fixed in 3% formaldehyde for 10 min, washed three times in PBS, and stained overnight at 37 °C in X-gal solution (1 mg/ml X-gal, 40 mM citric acid/Na phosphate buffer (pH 6.0), 5 mM $K_3Fe(CN)_6$, 5 mM $K_4Fe(CN)_6$, 150 mM NaCl, 2 mM $MgCl_2$).

## EdU cytochemistry

Cells plated on coverslips were cultured with 10 μM EdU for 24 h before being fixed in 4% formaldehyde for 10 min. Cells were washed in PBS, incubated in 100 mM Tris (pH 7.6) for 5 min, and then permeabilised in PBS + 0.1% Triton X-100 for 10 min. Cells were washed again in PBS and then placed onto a 50 μl drop of staining solution (2 mM Cu(II)$SO_4$, 4 μM Sulfo-Cy3-azide, 20 mg/ml $C_6H_7NaO_6$) on parafilm for 30 min. Cells were washed in PBS before incubating in 0.5 μg/ml DAPI for 5 min. Cells were washed again in PBS before mounting with mounting medium.

## Viability and apoptosis assays

For viability assays, ~10,000 cells were plated per well into 96-well plates. Drugs were added for the incubated period before replacing with fresh medium. One day later, viability was measured using the CellTiter 96 Aqueous One Solution Cell Proliferation Assay (Promega) according to the manufacturer's instructions. For Annexin V/PI flow cytometry analysis of apoptosis, drugs were added to ~400,000 cells per well in 6-well plates for indicated time points. Cells were harvested and stained with the Dead Cell Apoptosis Kit (ThermoFisher Scientific) according to the manufacturer's instructions. Samples were processed on a BD FACSCanto II and results were analyzed using Kaluza. Caspase 3/7 activity was measured using the Caspase-Glo 3/7 assay system (Promega) according to the manufacturer's instructions. Drugs were added to ~10,000 cells per well in 96-well plates for 4 h after which caspase-glo reagent was added. Caspase activity was measured 1 h later.

## siRNA transfection

Approximately 35,000 cells per well in 96 well plates were used for siRNA transfections. Cells were transfected with 20 pmols *Mcl-1* siRNA (sc-35877, Santa Cruz) or 20 pmols control siRNA (sc-37007, Santa Cruz) using the siRNA reagent system (Santa Cruz) according to the manufacturer's instructions. 24 h later, siRNA was removed and cells were treated with ABT-263 for 24 h, after which cell viability was measured as described above.

## Quantitative real-time PCR

Total RNA was extracted using the Isolate II RNA mini kit (Bioline). cDNA was synthesized using the High-capacity cDNA reverse transcription kit (ThermoFisher Scientific). cDNA amplification was carried out with the Sensi-Fast Probe Lo-Rox Kit (Bioline) and the Universal Probe Library system (Roche) in a LightCycler 480 Instrument II (Roche). Tubulin was used to normalize $C_t$ values. Primer and probe sequences are as follows:

*Bcl-w* – F: TGGATGGTGGCCTACCTG, R: CGTCCCCGTATAGAGCTGTG

*Bcl-w* probe – Sense: GCGGCTGG, Antisense: CCAGCCGC

*Mcl-1* – F: AAGCCAATGGGCAGGTCT, R: TGTCCAGTTTCCGAAGCAT

*Mcl-1* probe – Sense: GCAGGAAG, Antisense: CTTCCTGC

*p16* primer– F: GAGCAGCATGGAGCCTTC, R: CGTAACTATTCGGTGCGTTG

*p16* probe – Sense: CTCCAGCA, Antisense: TGCTGGAG

*p21* – F: TCACTGTCTTGTACCCTTGTGC, R: GGCGTTTGGAGTGGTAGAAA

*p21* probe – Sense: GGGAGCAG, Antisense: CTGCTCCC

*IL-6* – F: CAGGAGCCCAGCTATGAACT, R: GAAGGCAGCAGGCAA CAC

*IL-6* probe – Sense: CTGGGGCT, Antisense: AGCCCCAG

*MMP-1* – F: GCTAACCTTTGATGCTATAACTACGA, R: TTTGTGCG CATGTAGAATCTG

*MMP-1* probe – Sense: GGGAGAAG, Antisense: CTTCTCCC

*Tubulin* – F: CTTCGTCTCCGCCATCAG, R: CGTGTTCCAGGCAGT AGAGC

*Tubulin* probe – Sense: GCCTGCTG, Antisense: CAGCAGGC

*β-actin* – F: CCAACCGCGAGAAGATGA, R: CCAGAGGCGTACAGGG ATAG

*β-actin* probe – Sense: CAGCCTGG, Antisense: CCAGGCTG

mouse *cdkn2a (p16)* – F: AATCTCCGCGAGGAAAGC, R: GTCTGC AGCGGACTCCAT

*cdkn2a probe* – Sense: GAGGAGAG, Antisense: CTCTCCTC

mouse *Sox10* – F: ATGTCAGATGGGAACCCAGA, R: GTCTTTGG GGTGGTTGGAG

*sox10* probe – Sense: CAGAGCCA, Antisense: TGGCTCTG

## Immunofluorescence

Senescent melanocytes were treated with either DMSO or ABT-263 for 24 h. Then, cells were washed twice with PBS and fixed with 4% PFA for 10 min. Following two washes with PBS, cells were blocked in 5% normal goat serum (Sanquin) supplemented with 0.3% Triton X-100 for 1 h at room temperature and then they were incubated with primary antibody against Mcl-1 (1:800, Cell Signalling Technology, Ab#94296) diluted in 1% normal goat serum supplemented with 0.3% Triton X-100 overnight at 4 °C. The next day, cells were washed twice with PBS and incubated with goat anti-rabbit Alexa Fluor 488 (ThermoFisher Scientific, R37116) for 45 min at room temperature. Cells were washed in PBS and dH$_2$O before incubating in 2 µg/ml DAPI for 5 min. Cells were washed again in PBS before mounting with mounting medium. MCL-1 fluorescence was analyzed in >50 cells per condition and expressed as Corrected Total Cell Flourescence (CTCF). Using ImageJ, images were transformed to 8-bit, and cells were manually selected with a free drawing selection tool. The area, integrated density and mean gray value were measured in both the cells and in the background in each image. CTCF was calculated as Integrated Density − (Area of each selected cell · Mean fluorescence of the background reads).

## Immunoblotting

Protein lysates were obtained by scraping cells in RIPA lysis buffer (Abcam). Protein concentration was measured using the Pierce BCA protein assay kit (Thermofisher Scientific). Cell lysates were resolved on a 12% SDS-PAGE gel and transferred onto a nitrocellulose membrane (Bio-Rad). Membranes were blocked in 5% Milk in TBS-T (TBS with 0.1% Tween 20) or 5% BSA in TBS-T for 1 h at room temperature and then incubated overnight in primary antibody overnight at 4 °C. Antibodies used are: (1:000, Bcl-2, #15071, Cell Signalling Technology), (1:000, Bcl-xL, #2764, Cell Signalling Technology), (1:1000, Bcl-w, #2724, Cell Signalling Technology), (1:1000, Mcl-1, #94296, Cell Signalling Technology), (Mcl-1, sc-12756, Santa Cruz Biotechnology), (p-p70S6K, #9234, Cell Signalling Technology), (p70S6K, #9202, Cell Signalling Technology), (Vinculin, V9131, Sigma-Aldrich). After washing in TBS-T, membranes were incubated with peroxidase-conjugated secondary antibody (Dako) in 5% Milk in TBS-T for 1 h at room temperature. Membranes were washed again in TBS-T and proteins were detected using the Amersham ECL prime western blotting detection reagent (GE Healthcare) and the ImageQuant LAS 4000 mini imaging machine (GE Healthcare). Protein bands were quantified using ImageQuant software (GE Healthcare) and normalized to vinculin bands.

For analysis of BRAF$^{V600E}$ OE, adherent cells were washed once with DPBS and lysed with 8M urea containing 50 mM NaCl and 50 mM Tris-HCl, pH 8.3, 10 mM dithiothreitol, 50 mM iodoacetamide. Lysates were quantified (Bradford assay), normalized, reduced, and resolved by SDS gel electrophoresis on 4–15% Tris/Glycine gels (Bio-Rad, Hercules, CA, USA). Resolved protein was transferred to PVDF membranes (Millipore, Billerica, MA, USA) using a Semi-Dry Transfer Cell (Bio-Rad), blocked in 5% BSA in TBS-T and probed with primary antibodies recognizing β-Actin (Cell Signalling Technology, #3700, 1:4000, Danvers, MA, USA), BRAF$^{V600E}$ (Sigma-Aldrich, #SAB5600047, 1:1000). After incubation with the appropriate secondary antibody, proteins were detected using either Luminata Crescendo Western HRP Substrate (Millipore) or ECL Western Blotting Analysis System (GE Healthcare, Bensalem, PA).

## Immunostainings

Nevi sections were deparaffinised by placing them in xylene two times for 10 min each. Sections were then rehydrated by washing in serial grades of ethanol (100%, 95%, 80%, 60%) for 5 min each. Sections were then rinsed in dH$_2$O 3 times for 3 min each. Antigen retrieval was carried out by placing sections in Tris-EDTA buffer (10mM Tris Base, 1mM EDTA Solution, 0.05% Tween 20, pH 9.0) and heating in a microwave for 10 min. Sections were cooled for 30 min in antigen retrieval buffer and then rinsed three times in TBS + 0.025% Triton X-100 for 5 min each. Sections were blocked in 5% normal goat serum (Sanquin) in TBS for 1 h at room temperature. Sections were then incubated overnight in Bcl-w antibody (1:200, Proteintech, 16026-1-AP), Bcl-xl antibody (1:1000, Cell Signalling Technology, 2764T), Bcl-2-A1 antibody (1:1000, Abcam, ab45413), Bcl-2 antibody (1:200, Santa Cruz Biotechnology sc-7382), Mcl-1 antibody, (1:200, Abcam, ab32087), or MelanA antibody (1:1000, Abcam, ab210546) at 4 °C in 5% goat serum in TBS. Sections were rinsed three times in TBS + 0.025% Triton X-100 for 5 min each and then incubated in goat anti-rabbit Alexa Fluor 488 (ThermoFisher Scientific, R37116) for 1 h at room temperature. Sections were then rinsed three times in TBS for 3 min each and incubated in 1 µg/ml DAPI for 5 min. Sections were rinsed again three times in TBS for 3 min each before mounting slides with mounting medium.

Formalin-fixed paraffin-embedded human skin tissue sections from organotypic tissue were stained for MelanA/MART (NCL-L-MelanA, Leica Biosystems) and hematoxylin and eosin (Hematoxylin Gill III and Eosin (Cat# 3801540 and 3801600, Leica Biosystems)). Staining was performed following the manufacturer's protocol for high-temperature antigen unmasking technique for paraffin sections. Immunohistochemistry staining was performed by the University of Pennsylvania Skin Biology and Diseases Resource-based Center (SBDRC). Tissue section quantification was performed according to previous reports[54,55]. Briefly, 4× photomicrograph images of representative tissue sections were taken using the Keyence BZ-X710 (Itasca, IL, USA). Tiff files of the images were saved and transferred to FIJI (ImageJ). Images corresponding to a single specific color were then analyzed to determine the number of pixels in each sample and normalized to epidermal area. The numbers of pixels representing MelanA staining were normalized to the total amount of epidermal area.

## Polysome analysis

Approximately 10 million senescent cells were seeded in 15 cm plates and per condition, cells from two plates were pooled. Cells were treated with DMSO or ABT-263 for ~16 h after which cells were scraped on ice in ice-cold PBS containing 200 µg/ml cycloheximide (CHX) and centrifuged at 2000 rpm for 5 min at 4 °C. Cell pellets were then resuspended in 500 µl polysome lysis buffer (110 mM CH$_3$CO$_2$K, 20 mM Mg(CH$_3$COO)$_2$, 10 mM HEPES pH 7.6, 100 mM KCl, 10 mM MgCl$_2$, 0.1% NP-40, 2 mM DTT, 150 µg/ml CHX) and homogenized with a dounce homogenizer. Lysates were centrifuged at 3500 rpm for 10 min at 4 °C. Supernatants were then loaded onto 10–60% sucrose gradients and centrifuged at 38,000 rpm for 120 min at 4 °C. Gradients were fractionated and monosome, light polysome, and heavy polysome fractions were pooled. RNA was extracted from pooled fractions

by addition of 185 µg/ml proteinase K, 10 mM EDTA, and 1% SDS. Samples were briefly inverted and incubated at 42 °C for 30 min. Equimolar volumes of phenol:chloroform were added, vortexed, and centrifuged at full speed for 8 min at room temperature. The upper aqueous phase was separated and 1.4 ml 100% EtOH, 58 µl sodium acetate, and 1 µl glycol blue was added before mixing and incubating overnight at −80 °C. Samples were centrifuged at full speed for 30 min at 4 °C and the pellet was washed with 80% EtOH. The samples were centrifuged again at full speed for 10 min at 4 °C and the pellet was air dried. Pellets were dissolved in 10 µl RNase-free water and incubated at 60 °C for 5 min. cDNA was synthesized and amplified using protocols as described above. $C_t$ values of *Mcl-1* were normalized to β-*actin* and the percentage of *Mcl-1* mRNA in each fraction was calculated as reported[56].

### Cycloheximide chase assay
Senescent melanocytes treated with either DMSO or ABT-263 for 12 h, were then treated with 5 µg/ml cycloheximide. Protein lysates were obtained as described 30, 60, 120, and 240 min after the addition of cycloheximide and immunoblotted for Mcl-1 as described.

### Nascent protein synthesis
Senescent melanocytes treated with either DMSO or ABT-263 for 24 h, were subsequently depleted of Methionine by incubating the cells for 60 min in Methionine-free medium (21013-24, Gibco) without FBS. Then the cells were treated with 50 µM L-azidohomoalanine (Click-IT AHA, C10102, ThermoFisher Scientific) for 4 h, after which the cells were lysed in the following protein lysis buffer: 50 mM Tris-HCl pH8.0, 1%SDS, supplemented with protease inhibitors (cOmplete mini, 04693159001, Roche). Click-IT reaction was performed according to the manufacturer's protocol using Biotin alkyne (B10185, ThermoFisher Scientific). The proteins were dissolved in RIPA-buffer (ab156034, Abcam) supplemented with protease inhibitors (Roche). Biotinylated proteins represent the nascent proteins. The protein samples have been immunoprecipitated using magnetic beads (Dynabeads™ Protein G for Immunoprecipitation; 10003D; Invitrogen) and an anti-Mcl-1 antibody (#94296, Cell Signalling Technology), followed by an immunoblot for Biotin (hyb-8; ab201341, Abcam). Immunoblots for Vinculin (Sigma-Aldrich) and Mcl-1 (Cell Signalling Technology) were used as controls.

### Mice
All the mice were maintained in the central animal facility (CDP) of University Medical Center Groningen (UMCG) under standard conditions. All the experiments were approved by the Central Authority for Scientific Procedures on Animals (CCD – License #AVD105002015339) in the Netherlands. All mice were fed ad libitum (chow provided by Special Diets Services Cat# RM1 for breeding animals and RM3 for experimental animals) and housed in open cages.

Male *Tyr::CreER[T2] ;BRAF[CA/+]* mice[32] at 9 weeks of age were topically treated each day with 2 mg of tamoxifen (T5648, Sigma-Aldrich) onto shaven backs for 5 consecutive days. For topical administration of drugs, 1.3 mM ABT-263 and/or 0.27 mM S63845 was made up in a solution of DMSO/ethanol at a ratio of 1:6 and topically administered onto a small area of pigmented skin each day for 10 days. Mice were euthanized by cervical dislocation at the end of the experiment. Nevus density was calculated as described[38] using ImageJ software.

### Mass spectrometry analysis
Mice topically treated with ABT-263 and S63845 were euthanized and skin biopsies were harvested. Skin samples were sonicated and methanol supernatants were harvested after 12,000 rpm, 5 min centrifugation. Methanol supernatants containing drugs in skin was measured by Liquid chromatography–mass spectrometry (LC–MS).

### Human-engineered melanoma xenografts
Organotypic skin grafts were cultured in medium modified from previous methods[39,53,54]. Specifically, the Keratinocyte Growth Media (KGM) used for keratinocyte-only skin grafts was replaced with modified Melanocyte Xenograft Seeding Media (MXSM). MXSM is a 1:1 mixture of KGM, lacking cholera toxin, and Keratinocyte Media 50/50 (Gibco) containing 2% FBS, 1.2 mM calcium chloride, 100 nM Et-3 (endothelin 3), 10 ng/mL rhSCF (recombinant human stem cell factor), and 4.5 ng/mL r-basic FGF (recombinant basic fibroblast growth factor). Primary human melanocytes were transduced with lentivirus-carrying BRAF(V600E). Transduced melanocytes ($1 \times 10^5$ cells) and keratinocytes ($5 \times 10^5$ cells) were suspended in 80 µL MXSM, seeded onto the dermis, and incubated at 37 °C for 4 days at the air–liquid interface to establish organotypic skin. After 4 days, 2.5 µM ABT-263 and 20 nM S63845 were added to three biologic replicates of organotypics and three biologic replicates were treated with vehicle (DMSO) control. Total drug concentration remained under one percent of total media volume. The organotypic cultures incubated for three days +/− compounds when they were taken for histology.

### Statistics
Graphpad Prism was used for statistical analysis. Details are provided for each experiment in figure legends.

### Reporting summary
Further information on research design is available in the Nature Portfolio Reporting Summary linked to this article.

## Data availability
Source data for all images, raw data, and quantifications are provided in the source data file. Any additional information can be provided following contact with the corresponding author (m.demaria@umcg.nl). Source data are provided with this paper.

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

## Acknowledgements

We thank Mirjam Koster (UMCG) for helping with immunostaining, Herman J. Woerdenbag (University of Groningen) for help with topical treatments, Gilles Diercks (UMCG) for providing nevi biopsies, Hjalmar Permentier, Peter Hornatovich and the Interfaculty Mass Spectrometry Center (University of Groningen) for helping with measurements of drug penetrance and Christian Blank (NKI) for sharing the BRAF mice. This work was supported by grants from the Dutch Cancer Foundation (KWF

#10989 to M.D.), the Horizon2020 EU framework Program (to J.K. and M.D.), and from the China Scholarship Council (to C.G. and M.D.).

## Author contributions

Conceptualization: J.K. and M.De.; data curation: J.K., C.G., M.Do., E.F., W.J.F., T.W.R., and M.De.; formal analysis: J.K., C.G., M.Do., E.F., W.J.F., and M.De.; funding acquisition: M.De.; investigation: J.K., E.F., S.M.B.,W.J.F., T.W.R. and M.De.; methodology: J.K., C.G., E.F., M.Do., S.M.B., W.J.F., T.W.R., and M.De.; project administration: M.De.; resources: J.K., S.M.B., W.J.F., T.W.R., and M.De.; supervision: M.De.; validation: J.K., E.F., S.M.B., W.J.F., and M.De.; visualization: J.K. and M.De.; writing—original draft: J.K. and M.De.; writing—review & editing: J.K., W.J.F. and M.De.

## Competing interests

M.D. is founder, shareholder, and advisor for Cleara Biotech and an advisor for Oisin Biotechnologies. Neither Cleara Biotech nor Oisin Biotechnologies were involved in the study. The remaining authors declare no other competing interests.
