## [Peer Review File · Nature Communications]

Targeting anti-apoptotic pathways eliminates senescent melanocytes and leads to nevi regressionEditorial Note: This manuscript has been previously reviewed at another journal that is not operating a transparent peer review scheme. This document only contains reviewer comments and rebuttal letters for versions considered at Nature Communications.

REVIEWER COMMENTS

Reviewer #2 (Remarks to the Author):

This is now another revised version of the manuscript, in which survival mechanisms of senescent melanocytes in vitro and nevus cell nevi in vivo were investigated and therapeutically exploited.

The authors responded adequately to my often, I admit, demanding and tenacious questions about clarity of mechanistic findings claimed, model system chosen, selection of drugs followed up and translational implications concluded.

As a result, the authors made smaller changes regarding the data presentation (e.g. with reference to Mcl-1 levels in senescent melanocytes) and substantially improved the manuscript by a much more balanced, cautious and scientifically stimulating discussion of their findings.

I have no objections anymore and find this very interesting and highly relevant manuscript suitable for publication in Nature Communications.

Reviewer #3 (Remarks to the Author):

There are two comments left from the previous review:

1. Using IR instead of B-raf.

In the reply by the authors, they state that "we have added several experiments that cover the oncogene-induced senescence system." None of these is presented in Fig. 1. If there are experiments that show similar results with b-raf to what is shown for the IR-treated cells that would make Fig. 1 substantially stronger. I strongly recommend considering this. Including such data in Fig.1 would make the point way more convincing and improve the ms.

2. Drug penetration - Following the explanation by the authors it seems that the procedure was performed appropriately and therefore the drug does penetrate the tissue. I have no further comments on this point.

Reviewer #2 (Remarks to the Author):

This is now another revised version of the manuscript, in which survival mechanisms of senescent melanocytes in vitro and nevus cell nevi in vivo were investigated and therapeutically exploited.

The authors responded adequately to my often, I admit, demanding and tenacious questions about clarity of mechanistic findings claimed, model system chosen, selection of drugs followed up and translational implications concluded.

As a result, the authors made smaller changes regarding the data presentation (e.g. with reference to Mcl-1 levels in senescent melanocytes) and substantially improved the manuscript by a much more balanced, cautious and scientifically stimulating discussion of their findings.

I have no objections anymore and find this very interesting and highly relevant manuscript suitable for publication in Nature Communications.

Authors: We thank the reviewer for the comments and suggestions. Addressing their concerns has improved the quality and the scientific validity of the study.

Reviewer #3 (Remarks to the Author):

There are two comments left from the previous review:

1. Using IR instead of B-raf.

In the reply by the authors, they state that "we have added several experiments that cover the oncogene-induced senescence system." None of these is presented in Fig. 1. If there are experiments that show similar results with b-raf to what is shown for the IR-treated cells that would make Fig. 1 substantially stronger. I strongly recommend considering this. Including such data in Fig.1 would make the point way more convincing and improve the ms.

Authors: We thank the reviewer for this comment and request. We have now analyzed the expression of the genes bclw and noxa in Braf-induced senescent melanocytes. Similar to what observed for IR-induced senescent cells, the levels of bclw and noxa are upregulated suggesting consistent and stimulus-independent altered gene expression of selected anti- and pro-apoptotic genes in senescent cells. These data are now included as Figure 1j and 1q.

2. Drug penetration - Following the explanation by the authors it seems that the procedure was performed appropriately and therefore the drug does penetrate the tissue. I have no further comments on this point.

Authors: We are glad that our method satisfies the initial request and convince this reviewer about drug penetrance.